# Functional Morphology of the Antennae and Sensilla of *Coeloides qinlingensis* Dang et Yang (Hymenoptera: Braconidae)

**DOI:** 10.3390/insects13100907

**Published:** 2022-10-06

**Authors:** Cui-Hong Yang, Hua Xie, Zhi-Xiang Liu, Pei Yang, Ning Zhao, Bin Yang, Zong-Bo Li

**Affiliations:** 1Key Laboratory of Forest Disaster Warning and Control in Yunnan Province, Southwest Forestry University, Kunming 650224, China; 2College of Traditional Chinese Medicine, Yunnan University of Chinese Medicine, Kunming 650500, China; 3College of Life Sciences, Southwest Forestry University, Kunming 650224, China

**Keywords:** parasitic wasp, sensory organs, scanning electron microscopy, ultramorphology, sexual dimorphism

## Abstract

**Simple Summary:**

The wasp *Coeloides qinglingensis* (Hymenoptera: Braconidae) parasitizes insects that damage pine trees, so it is a potential biocontrol agent. Here, we used scanning electron microscopy to examine the fine morphology of the antennae of the adults, as well as the type, shape, and distribution of antennal sensilla. We detected sexual dimorphism in antennal flagellar length, and observed nine morphological types of antennal sensilla. Importantly, dome-shaped sensilla and sensilla auricillica are reported for the first time for *C. qinlingensis* in this study. Their shape differs from that of sensilla described in other parasitic wasp species. We discuss the potential functions of the sensilla of *C. qinlingensis* through comparisons with those of sensilla of other parasitic wasps, including braconid wasps that parasitize concealed insect hosts. This information provides a solid foundation for further studies on the chemical communication and behavior of C. qinlingensis.

**Abstract:**

*Coeloides qinlingensis* Dang et Yang, 1989 (Hymenoptera: Braconidae) is a biocontrol agent of several scolytid pine pests in Southwest China. We examined the fine morphology of the antennae of adult *C. qinlingensis*, as well as the type, shape, and distribution of antennal sensilla, via scanning electron microscopy. The antennae of female and male *C. qinlingensis* are filiform and comprise a scape, pedicel, and 31–36 flagellomeres. We detected sexual dimorphism in antennal flagellar length but not in the length of other subsegments. A total of nine morphological types of antennal sensilla varying in cuticular pore structure are present in both sexes, including nonporous types (sensilla trichodea, sensilla chaetica (2 subtypes), and sensilla coeloconica); apical pore types (sensilla basiconica and sensilla auricillica); and multiporous types (dome-shaped sensilla and sensilla placodea (2 subtypes)). Dome-shaped sensilla and sensilla auricillica are reported for the first time for *C. qinlingensis*, and their shape differs from that of sensilla in other parasitic wasps. The functional morphology of the sensilla of *C. qinlingensis* was compared with that of the sensilla of other parasitic wasps, including those that parasitize concealed insects. This information provides a foundation for further research on the chemical communication and behavior of *C. qinlingensis*.

## 1. Introduction

The parasitic wasp *Coeloides qinlingensis* Dang et Yang, 1989 (Hymenoptera: Braconidae) mainly parasitizes the larvae of several scolytid pine pests, including *Dendroctonus armandi* Tsai & Li, 1959, *Ips acuminatus* Wood & Bright, 1992, *Tomicus minor* Wood & Bright, 1992, *T. yunnanensis* Kirkendall & Faccoli, 2008, and *T. brevipilosus* (Eggers, 1929), all of which are notorious pests of coniferous trees in China [1,2,3]. The three aforementioned *Tomicus* species have been the main cause of the death of 1.5 million hectares of *Pinus yunnanensis* forests in Yunnan Province, Southwest China over the past 30 years [2]. Chemical pesticides, which are often used to control the spread of pests, are not effective against these pests because they remain hidden under pine bark on the shoot. Our field surveys have revealed that *C. qinlingensis* is a dominant parasitoid of these three *Tomicus* species, and the natural parasitism rates of *T. minor*, *T. yunnanensis*, and *T. brevipilosus* are 35.8%, 30.1%, and 21.5%, respectively; the highest rate of parasitism that has been observed to date is 67.1% (Li et al., unpublished data, also see [4]). The potential damage caused by *Tomicus* spp. and their population growth rate are significantly reduced after parasitization because of the non-reproductive and reproductive mortality of the parasitized host larvae [1,5]. Thus, this parasitoid wasp is a potential biocontrol agent for mitigating the spread of *Tomicus* spp. in southwest China.

The antennae of insects, especially hymenopteran parasitoids, contain various sensory organs (e.g., peg, hair, and plate) for detecting physical and chemical cues in complex ecological environments [6,7]. These senses allow parasitoids to execute a series of behaviors on the basis of the information received. Such behaviors include the colonization of appropriate habitats, host orientation, host recognition, courtship and mating, and avoidance of predators and other threats [7,8,9,10,11]. Like most parasitic wasps, *C. qinlingensis* adults use antennal sensilla to locate and evaluate the quality of habitats, hosts, and mates. After locating an appropriate host pine, female *C. qinlingensis* land on the surface of the bark and initiate a series of behaviors, including rapid walking while alternately drumming the antennae, similar to the sequence of behaviors that has been documented for other parasitoids [7,8,11,12]. The main difference between the oviposition events of *C. qinlingensis* and those of other parasitoids is the requirement for direct contact with the support surface via the last outwardly curved flagellomeres. During mate recognition, the mounting behavior of *C. qinlingensis* is triggered by antennal contact, especially via contact with the antennal flagellomeres, between female and male wasps. The antennae and sensilla of *C. qinlingensis* might play key roles in decision-making processes, including the process of host location, as well as partner recognition and acceptance. Studies on the functional morphology of antennal sensilla are critically important for enhancing our understanding of the behaviors of parasitic species [7,8,10,13].

Information on the antennal sensilla of *Coeloides* species is lacking. A study of the sensilla placodea (Sp) on the antennae of *Coeloides brunneri* Viereck, 1911 is the only study to date that has examined the antennal sensilla of a *Coeloides* species [14]. Furthermore, no studies have examined the type, morphology, and distribution of the antennal sensilla of *C. qinlingensis*. This is important information, given that this parasitoid depends on the antennae to receive cues that are used to make decisions. We studied the fine morphology of the antennae and sensilla of *C. qinlingensis* using scanning electron microscopy (SEM) as part of an ongoing effort to understand its mating and host location behavior. Specifically, in this paper we describe the type, morphology, location, number, and distribution of the different sensilla on the antennae of female and male *C. qinlingensis*. The functional morphology of sensilla was compared with that of sensilla in other parasitic wasps, especially braconid wasps that parasitize concealed insect hosts.

## 2. Materials and Methods

### 2.1. Insects

Yunnan pine (*P. yunnanensis*) timber (length: 40–50 cm, id: 22–25 cm) containing *C. qinlingensis* larvae was obtained from Jiulong Forestry Farm, Zhanyi District, Qujing, Yunnan Province, Southwest China (25°0′35″ N, 103°7′15″ E). The larvae were then transferred to the Laboratory of Forest Pest Control, Southwest Forestry University, and cultured in a 55 cm × 30 cm × 30 cm nylon bag at room temperature (23 ± 1 °C), with ca. 70% relative humidity, and a 16 h light: 8 h dark photoperiod, until adults emerged. Adults were placed in 50 cm × 50 cm × 50 cm insect-rearing cages and fed with 10% (*v*/*v*) honey solution. Finally, these parasitic wasps were identified on the basis of diagnostic characteristics, including the habitus, color, head, mesosoma, and veination of the forewing (Figure 1) [1,3].

### 2.2. Scanning Electron Microscopy

A total of 11 female and 11 male adults of *C. qinlingensis* were killed by fumigation with ethyl acetate. Sexes were identified following the descriptions provided in [1]. One antenna of each wasp was carefully excised from the antennal socket with microsurgical forceps (Rhino SW-11, Rhino Global Limted, Tokyo, Japan) under a Zeiss stereomicroscope (SteREO Discovery V20, Carl Zeiss, Gottingen, Germany) at 20× magnification. The antennae were first sonicated for 45 s in a 10% *w*/*v* saline solution containing detergent and then rinsed with 0.01 mol/L phosphate-buffered saline (pH 7.4) for 30 min. The clean antennae were fixed in 2.5% *w*/*v* glutaraldehyde for 4 h at the most. After fixation, antennae were dehydrated in an ethanol gradient (50%, 60%, 75%, 80%, 95%, and absolute ethanol) and then in acetone (100%) three times for 10 min before drieing in a critical point dryer (Quorum K850, Ashford, Kent, UK). Antennae were mounted with ventral, dorsal, and lateral sides uppermost on aluminum stubs with double-sided adhesive tape. The specimens were then sputter-coated with an automatic etching and coating system (Cressington 108, Chalk Hill, Watford, UK) and visualized under scanning electron microscopy (Sigma 300, Carl Zeiss Microscopy GmbH, Jena, Germany) at an accelerating voltage of 7 kV and a working distance of 4–15 mm. Micrographs of the antennae, antennal segments, and sensilla were obtained. The number and distribution of the sensilla on the antennae of both sexes were determined according to the descriptions of Renthal et al. [15]. Sensillar dimensions, including the length, width, and diameter of the sensilla base and socket, were measured using ImageJ 1.53k (htpp://imagej.nih.gov/ij, accessed on 15 June 2022, National Institutes of Health, Bethesda, MA, USA).

### 2.3. Sensilla Classification and Terminology

Antennal sensilla were classified on the basis of the external morphology visible in the micrographs, and the terminology followed that of Zacharuk [6], Gao, et al. [16], and Ahmed, et al. [17]. The description of the sensilla focused on morphological characteristics, such as the presence and location of pores. Variations in the length of single types of sensilla were not explored. In Ahmed et al. [17], 2 separate sensilla chaetica (Sch) on the antennal scape segment of *Macrocentrus cingulum* Brischke, 1882, were divided into two subtypes (CHA I and CHA II). In this study, Sch at the same location were not divided into subtypes, although some differences in shape and size were evident.

### 2.4. Data Analysis

The data were analyzed using the open-source software R 4.0.5 (https://cran.r-project.org/, accessed on 20 June 2022). Mean values for each sensilla type were calculated from at least 40 individual sensilla. The density of Sp was the total number within a 100-μm circumference and the density of surface pores was the number in 1-μm^2^ area [16]. The surface area of Sp was calculated using the method of Yang et al. [18]. Analysis of variance to detect differences between females and males was performed using a nonparametric Mann–Whitney U test because the data were not normally distributed (*p* < 0.05).

## 3. Results

### 3.1. Antennal Map of C. qinlingensis

The antennae of female and male *C. qinlingensis* are filiform and situated in a single membranous acetabulum (i.e., antennal socket), where they move freely in all directions (Figure 2A). The external morphology of female and male antennae is similar and consists of a basal scape, a pedicel, and a flagellum with 31–36 flagellomeres (Figure 2B and Figure 3). The number of flagellomeres is most often 36 in both females and males; the number of flagellomeres was 36 in 55.9% of all antennae examined (N = 34, 6 antennae of each sex examined by light microscopy, Figure 3). Thus, the subsequent description and comparisons of antennae and sensilla are based on flagella with 36 flagellomeres.

Generally, each antennal segment is a cylinder-shaped structure. The scape is the most proximal segment, which has an inconspicuous jointed segment called the radicula in the antennal socket (Figure 2A,B). There are no significant differences in the length and width of the scape between females and males (Table 1; length: Z = −0.492, *p* = 0.652; width: Z = −1.543, *p* = 0.133). The second segment is the pedicel, and there are no significant differences in the length and width of the pedicel between females and males (Table 1; length: Z = −1.018, *p* = 0.332; width: Z = −1.609, *p* = 0.116). The flagellum is the longest segment. Each flagellomere is similar in shape, but the first flagellomere has a strong protuberance apicoventrally; the second flagellomere normally flares apicoventrally, and the third flagellomere has a slight flare apicoventrally (Figure 2B inset). Flagellomeres are significantly shorter in females than in males (Table 1; Z = −6.116, *p* < 0.0001), but the width does not significantly differ between the two sexes (Table 1, Z = 0.813, *p* = 0.4174). Finally, there is slight sexual dimorphism in the whole antenna size between female and male *C. qinlingensis* (Z = −2.856, *p* = 0.003).

### 3.2. Sensilla Type and Its Morphology

The surface of the antennae, especially the antennal flagellum, is densely covered with various types of sensilla (Figure 2B). According to external morphology, presence or absence of pores, pore distribution, and the type of cuticular attachment (flexible or inflexible), nine types of sensilla were identified: sensilla trichodea (St), sensilla chaetica (Sch), sensilla basiconica (Sb), sensilla placodea (Sp), dome-shaped sensilla (Ds), sensilla auricillica (Sa), and sensilla coeloconica (Sco), with Sch and Sp each subdivided into two subtypes (Table 2). Additionally, cuticular pores (Cp) are present on the antennae. Generally, a sensilla with a flexible socket (Figure 4A) has a thin cuticular membrane, which functions as a connector between the antennal body and hair. This structure allows for movement at the base of sensilla. Conversely, some sensilla without flexible sockets (inflexible) are not entirely mobile (Figure 4B). Below, the types of sensilla and their fine morphology and cuticular pore structure are described in detail.

#### 3.2.1. Sensilla Trichodea (St)

St are slender, straight or slightly curved, ribbed hairs that vary in length (Table 2). They are aporous and arise from flexible cuticular sockets (Figure 4A and Figure 5). They range from 25.25 to 89.03 μm in length and from 2.55 to 6.86 μm in basal width, depending on their specific position on the antennae. Among all the sensilla types, St are the most abundant and are present on every antennal segment. They are particularly densely distributed on the antennal flagellomeres in both female and male wasps (Figure 6A,B, Figure 7A and Figure 8A,B; Table 3).

#### 3.2.2. Sensilla Chaetica (Sch)

Sch are stiff, straight, ribbed (subtype 1, Sch1), or smooth (subtype 2, Sch2) bristle-like structures, but they are stronger and thicker than St (Figure 5; Table 2). Sch are aporous and arise from a cuticle with a flexible cuticular socket (Figure 6C,D). Subtype Sch1 are elongated with pointed tips. The length of Sch1 ranges from 28.6 to 45.23 μm, and the basal width ranges from 1.55 to 3.50 μm. In terms of distribution, Sch1 are usually mixed with St on all antennal segments, but they are more commonly present on the front and lateral region of every segment in both sexes (Figure 6A,B and Figure 8A,B). Subtype Sch2 are small and short bristles (7.32–15.19 μm long, 1.17–2.55 μm wide at the base) with a wide socket (1.83–4.53 basal diameter; Figure 6D). They have sharp tips. These sensilla are visible dorsolaterally on the basal part of the scape (radicula) as well as the scape–pedicel joint in both sexes (Figure 6A,B).

#### 3.2.3. Sensilla Basiconica (Sb)

Sb are ribbed pegs that project slightly more perpendicular to the antennae than St and Sch described above (Figure 7A; Table 2). They range from 23.56 to 35.35 μm in length and 2.07 to 4.10 μm in basal width. The stem of these sensilla gradually curves at the 1/3 position. They end with a blunt tip, where 2-3 pores are located. Sb emerge from a flexible cuticular socket (Figure 7B). These sensilla are usually located at the distal part of every flagellomere (Figure 7A and Figure 8A,B).

#### 3.2.4. Sensilla Placodea (Sp)

Sp are sausage-like structures that are slightly elevated above the antennae surface, but there is no grooved gap separating them from the antennae surface (Figure 7A and Figure 8A,B; Table 2). There are two sub-types that differ in their external shape and the density of surface pores; Sp1 (Figure 8C,E) and Sp2 (Figure 8D,F). Sp1 are flat, smooth, with low-density pores, and are surrounded by a cuticular ridge parallel with the plate. They range from 108.99 to 127.25 μm in length and from 3.87 to 6.91 μm in width. Generally, Sp1 are arrayed parallel to the antennal axes and are located between rows of St and Sch1. They are present on all the antennal flagellomeres, except for the first flagellomere in both sexes (Figure 8A,B). Sp1 are relatively sparsely distributed, with only 4–7 individuals on the flagellomeres 2–13, but they are significantly more abundant on the rest of flagellomeres (9–12 individuals) (Z = −3.782, *p* < 0.0001; Figure 9). Sp2 are concave and rough, have high-density pores, and have cuticular ridges on the plate surface. The sensilla plate is about 83.15–119.51 μm in length and 3.80–7.68 μm in width. The arrangement of Sp2 is similar to that of Sp1, but the former is usually located on the lateroventral side of every flagellomere (Figure 8B, Figure 10A and Figure 11). Compared with Sp1, Sp2 are more sparsely distributed, with only 2–4 per flagellomere. The numbers of Sp2 are almost equal among the flagellomeres (Figure 9).

#### 3.2.5. Sensilla Auricillica (Sa)

Sa on *C. qinlingensis* are ear-expanded structures that slightly protrude from the antennal surface (Table 2). The outward-facing aspect of Sa is located in a flexible cuticular socket and is concave, with a rough surface and a round pore in the center of the ear (Figure 10A,B). Sa ear has a large diameter at the top (1.42–1.56 μm) and a smaller diameter at the bottom (0.72–0.76 μm). These sensilla are mainly located on odd-numbered flagellomeres 15–33 with only one per flagellomere (Table 3). A few sensilla are also present on other flagellomeres, e.g., flagellomeres 3, 5, or 7, but only one is present on each flagellomere.

#### 3.2.6. Dome-Shaped Sensilla (Ds)

Ds consist of a cupola-like peg with a blunt tip, which is clearly wider than it is long (Figure 4B; Table 2). The wall of Ds is a thick, multiple-layer structure with slightly longitudinal grooves and some pores (Figure 11 inset). These sensilla are distributed only on the ventral side of flagellomeres 1–4 and usually in groups of 24–124 individuals (Figure 11). Ds are least abundant on flagellomere 1 and 4 (24–56) and most abundant on flagellomere (83–124 in females and males, respectively) (Table 3).

#### 3.2.7. Sensilla Coeloconica (Sco)

Sco are short pegs in the pit sensilla that have a terminal finger-like projection and are embedded in an inflexible socket (Figure 12A; Table 2). Sco obviously grow over the cavity and range from 2.76 to 4.81 μm in length and from 0.85 to 2.07 μm in basal width. This type of sensilla is sparsely distributed on the lateroventral side of even-numbered flagellomeres 8-36 (Figure 7A; Table 3). There is only one Sco on each flagellomere.

#### 3.2.8. Cuticular Pore (Cp)

The Cp look like small holes and are randomly distributed on every flagellomere of both sexes (Figure 8A,B, Figure 10A, Figure 11 and Figure 12B; Table 2). There are often 2–3 Cp per flagellomere, but flagellomeres 10–20 have 5–8 each (Figure 8A,B; Table 3). The Cp diameter ranges from 0.41 to 0.90 μm.

### 3.3. Number and Distribution of Sensilla on Both Sexes

All types of sensilla are present on the antennae of female and male *C. qinlingensis* (Table 3). They show the same distribution pattern on the antennae of both sexes. However, the number, size, and density of some types of sensilla differ significantly between the sexes (Figure 9 and Figure 13; Table 3 and Table 4). The numbers of some types of sensilla (St, Sch1, and Sch2) on the scape and pedicel do not differ significantly between the two sexes (*p* = 0.063–1.0). However, there are significantly larger numbers of St, Sch1, Sp1, and Ds on the antennal flagellum in males than in females (St: Z = −2.611, *p* = 0.0079; Sch1: Z = 2.578, *p* = 0.016; Sp1: Z = −2.402, *p* = 0.015; Ds: Z = −2.677, *p* = 0.0079; Table 3). In terms of sensilla size, Chs1 and Sco are significantly larger in males than in females (Chs1 length: Z = −4.8594, *p* < 0.0001, width: Z = −3.002, *p* = 0.0024; Sco length: Z = 2.030, *p* = 0.042, width: Z = −3.123, *p* = 0.0016; Figure 13A). Additionally, Sb are significantly longer in males than in females, but the width does not differ significantly between the two sexes (Figure 13B). The length of Sp2 are not different between females and males, but the width of Sp2 is greater in males than in females. There are more Sp1 on every flagellomere in males than in females (*p* = 0.016–0.032; Figure 9A), but the number of Sp2 on every flagellomere does not differ significantly between females and males (*p* = 0.14–0.55; Figure 9B). The density of Sp is particularly relevant because it may explain differences in olfactory sensitivity. The densities of Sp1 and Sp2 on antennae are not significantly different between females and males (Table 4). However, males have a significantly greater pore surface and total surface area, so there are more Sp1 and Sp2 in total in males than in females (all *p* < 0.0001; Table 4).

## 4. Discussion

In this study, we studied the external morphology and the types, position, number, and distribution of sensilla in female and male *C. qinlingensis*. As in most insects, the antennae of *C. qinlingensis* typically comprise a scape with a pedicel inserted into the antennal socket, a short pedicel, and a flagellum composed of 31–36 flagellomeres. Onagbola and Fadamiro [19] suggested that the radicula is a separate segment with different sensilla associated with distinct intersegmental regions. However, the pedicel of *C. qinlingensis* antennae has an unseparated region that connects the scape and radicula. It can be clearly seen ventrally; a faint gap can also be observed dorsolaterally (Figure 6A). Thus, this can be regarded as a scape segment on the antennae of *C. qinglingensis*, and is consistent with the generally accepted structure and function of a scape segment [20]. Additionally, specialized derived characteristics, including the strong protuberance on flagellomeres 1–3, have been observed in 11 *Coeloides* species that mainly parasitize the larvae of some concealed forest pests of Scolytidae, Curculionidae, and Buprestidae (Coleoptera) [1]. The distribution of dome-shaped sensilla in *C. qinlingensis* might permit adjustments of the positioning of the antennae to facilitate mate recognition; parasitoids generally flick the antennae against those of potential mates [10]. Additional studies on mating behaviors using high-definition video are needed to confirm the hypothesis that strong protuberance is a morphological adaptation.

### 4.1. General Distribution of Antennal Sensilla

The distributions of the different types of sensilla in this study do not differ markedly from those observed in braconid wasp and other hymenopteran parasitoids [7,9,10,13,16,17,21,22]. Most sensilla are located on each side of antennal segments and are most abundant on the flagellum. The three exceptions to this pattern include Sa and Ds, which are mainly located on the ventral side of flagellomeres, and Sco, which occur only on the lateroventral side of flagellomeres. Six of these sensilla types correspond to those described in wasp species in Braconidae [7,8,10,13,16,17,21,23,24], Ichneumonidae [25], Pteromalidae [18,19], Chalcidoidea [26], Platygastroidae [9], Eulophidae [27,28], Bethylidae [29], Cynipidae [22,30], Agaonidae [18], and Aphidiidae [31]. The external shape of other sensilla types (at least Sp2 and Sa) varies, and differs from those described for the braconid wasp *B. vulgaris* Ashmead, 1894 [21] and *C. brunneri* [14]. There is only a single type of Sb on the antennal flagellomere of *C. qinlingensis*. This might facilitate specific behaviors that permit parasitoids to locate larvae via the contact chemicals that they release [10,21].

### 4.2. Morphology and Mechanoreception of Sensilla

Mechanosensilla allow insects to sense mechanical movement generated by contact with other objects, air currents, sound, gravity, or pressure [20]. In *C. qinlingensis*, the hairlike St are numerous, occur widely on the antennae, and are morphologically similar to those in the braconid species *Microplitis croceipes* (Cresson, 1872) [13], *Cotesia glomerata* (Linnaeus, 1758) [23], *M. pallidipes* Szepligeti, 1902 [16], *M. cingulum* [17], and *B. vulgaris* [21]. Transmission electron microscopy (TEM) has revealed that St are a nonporous type of sensilla. In general, nonporous sensilla are assumed to be mechanoreceptive sensilla and not chemoreceptive sensilla. Consistent with this, the results of previous studies have shown that sensillar function is determined by the number and position of pores [23,32,33]. St show great variation in shape and size and densely cover the antennae, except on the basal segments (Table 2). The socket-like base of St is mobile [20], and it can sense mechanical stimuli when touched, moved, or distorted. Some St can receive airflow [34]. Thus, St in *C. qinlingensis* might be air-sensitive receptors that receive antennal mechanosensory information and aid decision-making during flight [8,21,34].

Ribbed Sch1, which have a nonporous surface, have been identified in various parasitic wasp species, including the eulophid *Tetrastichus hagenowii* (Ratzeburg, 1852) [28] and *Quadrastichus mendeli* Kim & La Salle, 2008 [35], the pteromalid *Philotrypesis longicaudata* Mayr, 1906, and *Sycoscapter roxburghi* Joseph, 1957 [18]; in those studies, they were referred to as aporous socketed hairs, sensillum chaeticum C1, and sensilla chaetica type 1, respectively. These sensilla are less numerous than St, but they are more robust and longer than the other types of sensilla. They might permit parasitic wasps to recognize the soft membranes or sclerites of hosts and to determine the position of their antenna with respect to their surroundings [8]. Sch1 in *C. qinlingensis* has no terminal pore and are stiffer than other types of sensilla. They also tend to be the first sensilla to make contact with the substrate. Thus, mechanical cues are directly detected from the exterior of the concealed host.

Smooth Sch2 widely occur in parasitoid species and many other insects [7,20,22,30]. These sensilla are always located on the articulation between the head and scape and between the scape and pedicel. In other studies, Sch2 have been referred to as aporous type 4 sensilla trichodea [19], variable-sized sensilla chaetica [16], nonporous sensilla chaetica type I and II [17], and Böhm sensilla [31,35]. On the basis of their external morphology, Sch2 in some parasitic wasps are proposed to function as mechanoreceptors or proprioceptors, which are capable of detecting the position and movement of the antennae through a stimulus produced by antennal distortion [6,10,17,20,35]. Similarly, Sch2 in *C. qinlingengsis* is thought to be a proprioceptor that can detect and transmit diverse mechanical stimuli from antennal movement [8,10,11,21].

Sa also perform another mechanical function. These sensilla are typically not observed on the antennae of parasitic wasps; this is the first study to confirm their presence on the antennae of *C. qinlingensis*. Sa are usually found on the antennal scape and pedicel, as well as on the flagellar segments of some species of Lepidoptera [36], Hemiptera [37], Trichoptera [36], and Coleoptera [38]. The human ear-shaped Sa in *C. qinlingensis* are slightly similar to the labial palps of the trichopteran *Limnephilus marmoratus* Curtis, 1834 [39], but completely different from the rabbit ear-shaped Sa reported in previous studies. This, coupled with their multiparous surfaces, suggests that these sensilla perform chemosensory roles [36,39]. The morphology, structure, and distribution of Sa observed in this study suggest that they play a mechanosensitive role; however, a single micropore is present at the center of the ear-shaped structure. The opening of this micropore is larger than that on the surface pores of sensilla observed in previous studies [13,36,39], and it resembles an ear canal that might be responsible for funneling mechanical waves. Danci et al. [40] indicated that male *Pimpla disparis* Viereck, 1911 can detect the presence of acoustic and vibratory cues of sexual partners to track the developmental progress of future females inside host pupae. We note that *C. qinlingensis* typically stops and arranges itself parallel to the host with the ventral antennae bearing Sa during the host location process, which suggests that these sensilla might play a role in sensing sound and vibratory signals and identifying appropriate host larvae [5]. This behavior has also been observed in *O. concolor* Szepligeti, 1910 [8] and *B. vulgaris* [21], but these two parasitoids do not possess Sa. Thus, Sa in *C. qinglingensis* appear to aid wasps in locating their concealed host insects under the bark.

### 4.3. Morphology and Chemoreception Functions of Sensilla

The characteristic features of chemosensilla are the cuticular porous structures that allow chemical molecules to stimulate receptors within the sensillar lymph cavity [20]. These sensors in insects can be classified as multiporous olfactory sensilla or uniporous gustatory sensilla, according to the number and location of cuticular pores [9,32]. In *C. qinlingensis*, Sb with a terminal pore are likely a uniporous sensilla type that has been described as fluted Sb in the braconid *Cardiochiles nigriceps* Viereck, 1912 [25], tip-pore St in *C. glomerata* [23], uniporous Sch in the pteromalid *P. cerealellae* (Ashmead, 1902) [19], uniporous pitted sensilla in egg parasitoids [41], and uniporous gustatory sensilla in many parasitic wasps [18]. They protrude conspicuously over the other sensilla, suggesting that these organs have a gustatory function when wasps make contact with the surface of an object with the apical flagellomere of antennae, as has been observed in many hymenopterans [8,21]. In many TEM studies, typical Sb are non-perforated, thick-walled pegs inserted into inflexible or flexible sockets, with 3–10 innervated neurons with unbranched dendrites [6,13,16,32,35]. Furthermore, a few dendrites extend along the length of the peg and end in the tubular body, which is indicative of a mechanoreceptive function [27,32]. Sb in *C. qinlingensis* are likely mechano- and gustatory receptors, as described in the leafminer parasitoid *Sympiesis sericeicorniis* (Nees, 1834) [27]. In addition, Sb occur on the antennae of all insects, including parasitic wasps, and exhibit the same distribution, especially on the terminal flagellum, which might reflect convergent functional evolution of drumming behavior during food, host, and mate discrimination [7,13,20,21,30].

The multiparous Sp in *C. qinlingensis* are elongated plates that have been identified in nearly all parasitic wasps [7,13,14,26,30,42]; two exceptions are the bethylid *Scleroderma guani* Xiao et Wu, 1983 [29] and the agaonid *Eupristina* sp. (roughly oval plates invaginated into a pit) [43]. The multiple cuticular pores of these sensilla indicate that they are olfactory receptors that respond to chemical cues from plant volatiles and pheromones in the early stage of host selection and mate recognition [8,10,13]. Because of their high morphological similarity, the two Sp subtypes in *C. qinlingensis* are assumed to be olfactory organs; furthermore, these two Sp subtypes have been confirmed to be olfactory organs in the braconid *M. pallidipes* [13]. However, Sp in *C. qinlingensis* exhibit a few notable morphological differences, including the presence of an external encircling groove and a plate surface ridge, as well as the density of pores (Figure 6C–F), which might be directly related to the degree of functional specificity in their search for host insects and mates [14,21,26,42]. Whether the morphological differences in *C. qinlingensis* reflect differences in functional specificity requires further verification using single-sensor recording techniques.

Ds in *C. qinlingensis* have not been detected on the antennae of parasitic wasps and other insects, although sensilla with a similar name have been described in some elaterid beetles [44]. Their external morphology is similar to that of organs present on the apical segment of the ovipositor in a few parasitoids [45], but multiple pores are present on the cone wall. Ds in this species likely have olfactory functions [9,32]; thus, they are not thermoreceptors or thermo-hygroreceptors [44]. In *C. qinlingensis*, Ds are only distributed on the lateroventral region of flagellomeres 1–4 with their tips above the antennal cuticle. This concentrated distribution suggests that they play a role in antennal contact between the sexes during pre-mating courtship. Therefore, they might facilitate the detection of sex pheromones over short distances during the process of sex recognition, given that males begin to emit secretions after their antennae-bearing Ds touch those of their mates [10].

### 4.4. Morphology and Thermo-Hygroreception of Sensilla

The thermo-hygroreceptive function appears to be restricted to Sco in *C. qinlingensis*. The main features of Sco are as follows: First, each Sco consists of a terminal projected peg located in either a deep tube or a pit; these Sco closely resemble those that have been observed in other parasitic wasps [7,17,22]. Braconid wasps are thick-walled, nonporous, and innervated by three neurons [16,32]. Second, Sco are less numerous than are mechanosensilla, and often occur singly [6,7,22], consistent with our observations in this study. Third, Sco are always hidden under the surface of the cuticle, and the peg extends from the pit perpendicularly. This might conserve space on the antennal cuticle or protect the sensory peg against environmental temperature fluctuations or water loss [20,32].

### 4.5. Cuticular Pores on the Flagellomeres and Their Function

The Cp described in this study are similar to the glandular outlets on the antennae of the cynipoid *Aylax minor* Hartig, 1840 [46]; the three braconids *Alysia alticola* (Ashmead, 1890), *Aphaereta pallipes* (Say, 1829), and *A. ervi Haliday, 1834* [10]; and those observed in species of the genus *Aphelinus* Dalman [47]. In previous studies, porous structures have been proposed to play a role in secretion rather than sensation, given that paste-like secretions have been shown to ooze from the pores upon antennal contact between the sexes [10]. Antennal contact with female antennae is generally made along the lateroventral portion of the antennal flagellum, especially from the base to the mid flagellomeres, during courtship. The behavior of *C. qinlingensis* is similar to that of *Pimpla turionellae* (Linnaeus, 1758) and *A. ervi*, and the flicking of their antennae-bearing Cp is essential for mate acceptance [10,48]. Behavioral and functional morphological observations indicate that the Cp in *C. qinlingensis* are glandular organs, although secretions around the pores were not observed in this study (Figure 7A,B, Figure 8A and Figure 10B). The males of many parasitic hymenopterans produce sex pheromones through Cp. Previous studies reported that Cp occur exclusively on the antennae of males [48,49]. However, the Cp in *C. qinlingensis* are also present in females, but with a slightly lower abundance than in males (Table 3). Quicke [49] suggested that the braconid parasitic wasps bearing dorsal glands could engage in other forms of chemical communication through air-borne or substrate-born pheromones. These glands might also play a key role in mediating contact recognition during mate encounters in the long or short range. Additional TEM observations are needed to clarify the significance of functional morphological differences and the role of chemical communication in mate selection.

### 4.6. Sexual Variability in the Antennae and Key Sensillar Equipment

In *C. qinlingensis*, the antennae are significantly longer in males than in females, consistent with the results of several previous studies on parasitic wasps [13,16,17,19,23,24,31,33,45]. In a few species, such as *C. nigriceps* and *Campoletis sonorensis* (Cameron, 1886) [25], as well as *Trichospilus pupivorus* Ferri è re, 1930 [50], the antennae are longer in females than in males. Although the number of flagellomeres is the same in male and female *C. qinlingensis*, the flagellomeres are always longer and wider in males than in females. This increase in the length and width of the flagellomeres in males suggests that bearing larger sensory organs might have an adaptive function.

Significant differences between the sexes were observed for five types of antennal sensilla: St, Sch1, Ds, Sp1, and Sp2. Both St and Sch1, the most widespread mechanosensilla, are significantly more numerous in *C. qinlingensis* males than in females, as also reported for braconid *M. croceipes* [13,25,33], the pteromalid *P. cerealellae* [19], and the aphidiid *Aphidius gifuensis* Ashmaed, 1906 [31]. The opposite pattern was observed for the antennae of *C. marginiventris* (Cresson, 1865), in which St are more numerous in females than in males [33]. Sex-specific differences in sensilla are common in hymenopteran parasitic wasp species. However, the causes of these sex-specific differences among parasitoids have not yet been investigated. The sex-specific differences in soft St and stiff Sch1 suggest that they are nonporous sensilla with mechanosensitive functions in insects. Sexual differences in the sensilla of *C. qinlingensis* might aid their flight and drumming behaviors and thus their ability to locate hosts. To our knowledge, this is the first study to document Ds in both sexes. The lateroventral position of Ds on flagellomeres 1–4 suggests that they might play a role in processing information associated with contact pheromones because antennal contact is the first step in female recognition and acceptance [10,48].

Perhaps the most intriguing finding of our study is the sexual variability in Sp. The total number of Sp per flagellomere, with the exception of solitary flagellomeres, is higher in males than in females in *C. qinlingensis*; thus, there are more Sp on male antennae than on female antennae (female Vs male = 368.4 ± 20.9 Vs 431.0 ± 29.2, *p* = 0.016). These results are consistent with the findings of previous studies on *C. brunneri* [14] and other parasitic wasps [13,16,17,18,19,21,24,25,30,31,33,34,36,45,50]. These results are also inconsistent with the findings of a few studies of the antennal sensilla of *C. marginiventris*, *C. rubecula* (Marshall, 1885), and *C. glomerata*, in which Sp are more abundant in females than in males [23,33]. Thus, the longer antennae length in males than in females does not explain the greater number of Sp in *C. qinlingensis* males than in females. Male parasitic wasps typically have more olfactory sensilla than do females because the male antennae of many species are adapted to sense sex pheromones emitted from females; females use their antennae to sense induced volatile semiochemicals from plants attacked by host herbivores. Indeed, some studies have demonstrated that Sp play a specific role in the detection of either herbivore-induced plant volatiles (HIPVs) or sex pheromones in parasitic wasps [13,33]. Although no sex pheromones have been identified in *C. qinlingensis*, the number, type, and distribution of Sp (i.e., more abundant on mid flagellomeres (Figure 9), as well as pre-courtship behavior induced by antennal contact in both sexes of *C. qinlingensis*, suggest that the effects of sensilla on behavior vary. In *C. qinlingensis*, the two Sp subtypes, Sp1 and Sp2, are distributed along the full length of each flagellomere and significantly differ in their pore density, surface pores, and total surface area between the sexes (Table 4). This might explain the sensitivity and putative functions of Sp1 and Sp2. Another study found that Sp1 is more abundant on the mid flagellomeres [42]. Sp1 with low pore density are more numerous in males than in females, but the number of high-pore-density Sp2 is similar in male and female *C. qinlingensis*. In light of the above, we propose the following two hypotheses: (1) the high abundance and low pore density of Sp1 in *C. qinlingensis* males are likely involved in the sensation of sex pheromones, and thus, should have higher sensitivity to sex pheromone components than to HIPVs; and (2) the low-abundance and high-pore-density Sp2 are likely involved in the detection of host-related volatiles in both sexes of *C. qinlingensis*, and thus, should have higher sensitivity to HIPVs than to sex pheromone components. Future research is needed to test the above hypotheses regarding the roles of Sp1 and Sp2 in the detection and sensation of different volatiles identified from *C. qinlingensis* and their attacked plants.

## 5. Conclusions

Using SEM analyses, we examined the fine morphology of the antennae of the larval parasitoid *C. qinglingensis*, as well as the type, shape, and distribution of antennal sensilla. We assumed that the function of antennal sensilla links to their morphology, especially the presence of cuticular pores, and to wasp behaviors. The nonporous types of sensilla, including St, Sch1, Sch2, and Sa, function as mechanosensilla and can sense mechanical stimuli generated by contact with other objects, air currents, or sound. The sensilla with apical pores, such as Sb, have double functions as a mechanosensilla and multiparous. The multiparous types of sensilla, including Sp1, Sp2, and Ds, can identify pheromones and host volatiles at the long or short range. Sco have thermo-hygroreceptive functions, and Cp are glandular organs that secrete contact pheromones during courtship. These assumptions need to be verified by further studies using TEM and single sensillum recording. This study is part of an ongoing effort to understand mating and host location behavior associated with different sensors and chemical communications.

## Figures and Tables

**Figure 1 insects-13-00907-f001:**
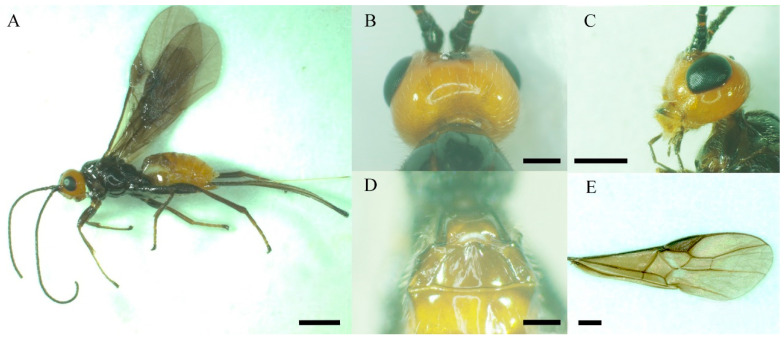
Diagnostic characteristics of adults of *C. qinlingensis*. (**A**) Habitus. (**B**) Head, dorsal view. (**C**) Head, lateral view. (**D**) Mesosoma, dorsal view. (**E**) Venation of right forewing. Scale bar: A = 1 mm μm; B and D = 0.2 mm; C and E = 0.5 mm.

**Figure 2 insects-13-00907-f002:**
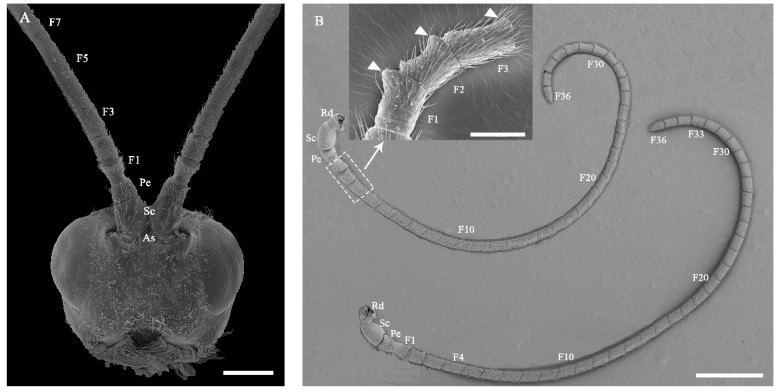
Head and antennae of adult *C. qinlingensis*. (**A**) The antenna base is situated in a single socket-like area of the head wall. (**B**) Whole antennae of female (**top**) and male (**bottom**) wasps. The inset shows strong protuberances when viewed apicoventrally (triangular arrow). As, antennal socket; Rd, radicula; Pe, pedicel; Fl, flagellum; F1–F36, flagellomere number. The same acronyms and abbreviations are used in subsequent figures. Scale bar: A = 200 μm; B = 500 μm, Inset = 100 μm.

**Figure 3 insects-13-00907-f003:**
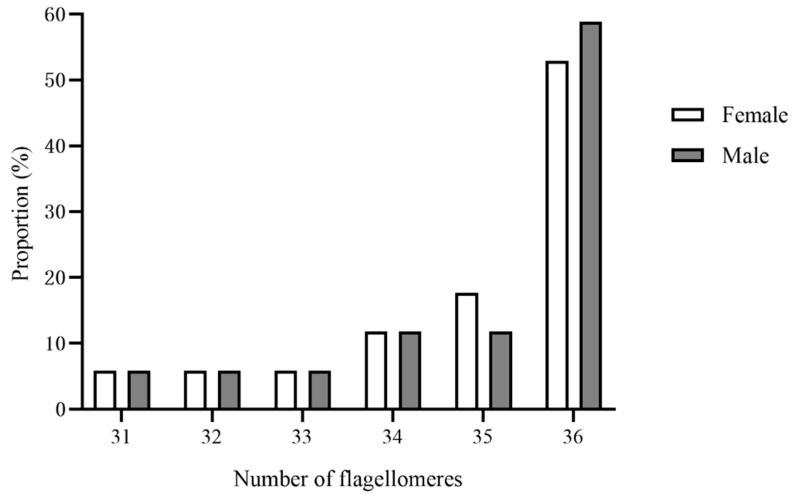
Proportion of total number of antennal flagellomeres in female and male *C. qinlingensis* (*n* = 34).

**Figure 4 insects-13-00907-f004:**
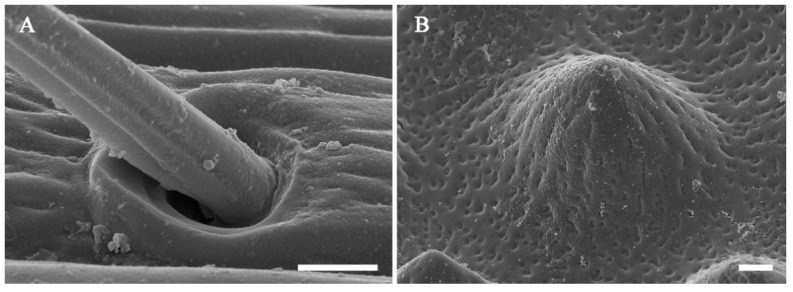
Type of sockets of the antennae of *Coeloides qinlingensis*. (**A**) Flexible socket at the base of sensilla trichodea. (**B**) Inflexible socket of dome-shaped sensilla. Scale bar: A and B = 1 μm.

**Figure 5 insects-13-00907-f005:**
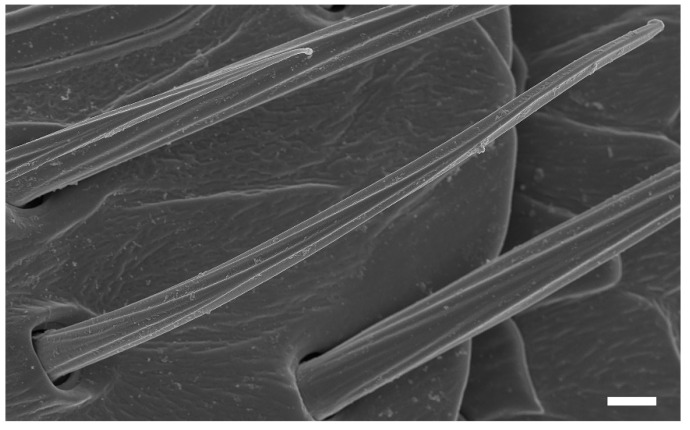
Characteristic ribbed hairs of sensilla trichodea emerging from flexible cuticular sockets. Scale bar = 2 μm.

**Figure 6 insects-13-00907-f006:**
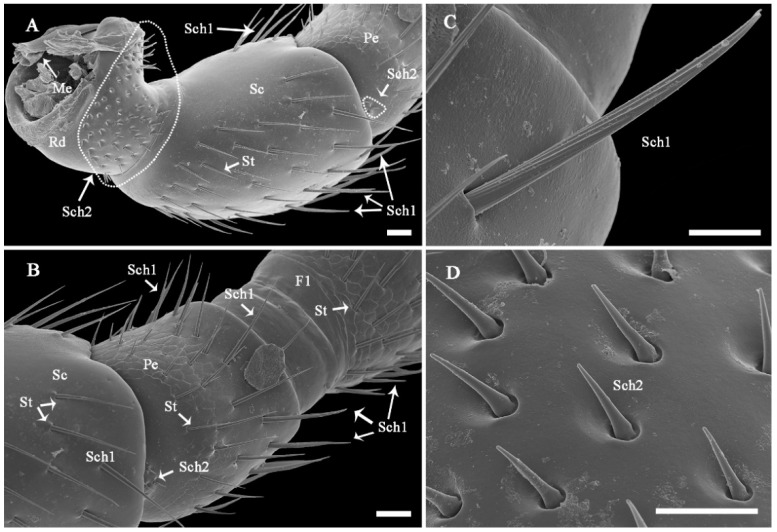
Fine morphology and distribution of antennal sensilla on antennal bases of *C**. qinlingensis*. (**A**) Lateral-dorsal view of the scape with a pedicel. The dotted circle shows the concentrated distribution at the radicula. (**B**) Ventral view of the pedicel. (**C**) Characteristic ribbed bristle of sensilla chaetica 1 located at a flexible socket located close to the pedicel tip. (**D**) Smooth bristle of sensilla chaetica 2 on the lateral side of the radicula. Me, membrane; Sch1-2, sensilla chaetica 1 and 2; St, sensilla trichodea. The same acronyms and abbreviations are used in the following figures. Scale bar: A and B = 20 μm, C and D = 10 μm.

**Figure 7 insects-13-00907-f007:**
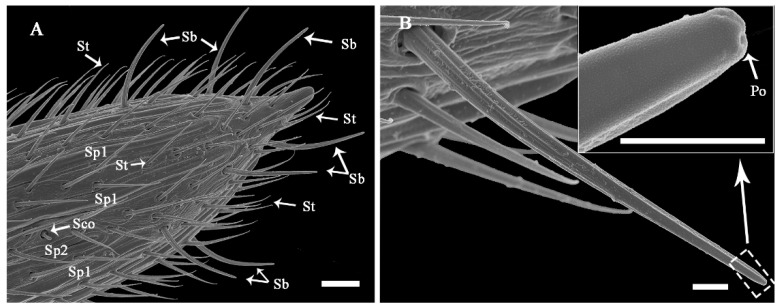
Fine morphology and distribution of antennal sensilla on the tip of the flagella of *Coeloides qinlingensis*. (**A**) Lateral view of the last flagellomere. (**B**) Characteristic ribbed peg stemming from a flexible socket at the flagellar tip. The inset shows a few pores at the end of the peg. Sb, sensilla basiconica; Sp1-2, sensilla placodea 1 and 2; Sco, sensilla coeloconica, Po, Pore. The same acronyms and abbreviations are used in subsequent figures. Scale bar: A = 10 μm; B = 2 μm, inset = 200 nm.

**Figure 8 insects-13-00907-f008:**
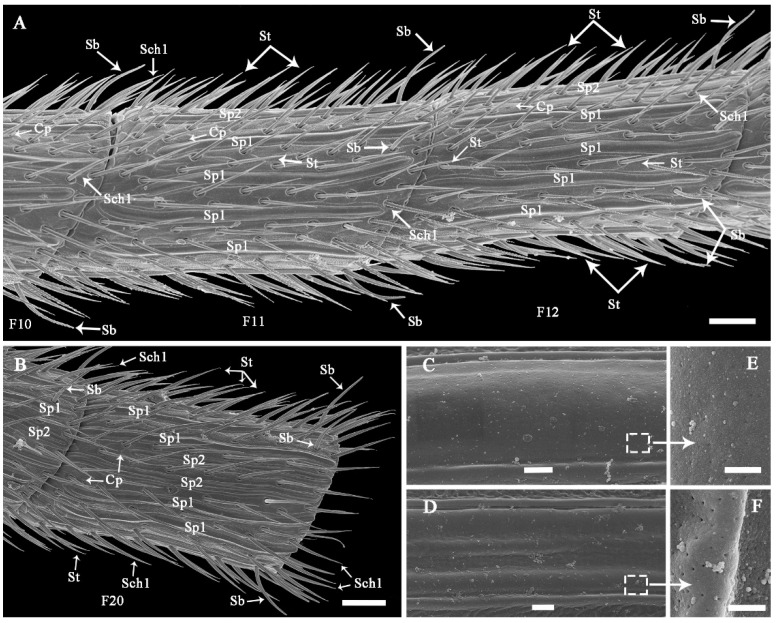
Fine morphology and distribution of antennal sensilla on mid flagellomeres of *Coeloides qinlingensis*. (**A**) Lateral view of the mid flagellomeres. (**B**) Lateral view of flagellomere 20. (**C**) Smooth surface of sensilla placodea 1. (**D**) Concave surface of sensilla placodea 2. (**E**) Low-density pores on sensilla placodea 1. (**F**) High-density pores on sensilla placodea 2. Cp, cuticular pore. The same acronyms and abbreviations are used in subsequent figures. Scale bar: A and B = 20 μm; C and D = 1 μm; E and F = 200 nm.

**Figure 9 insects-13-00907-f009:**
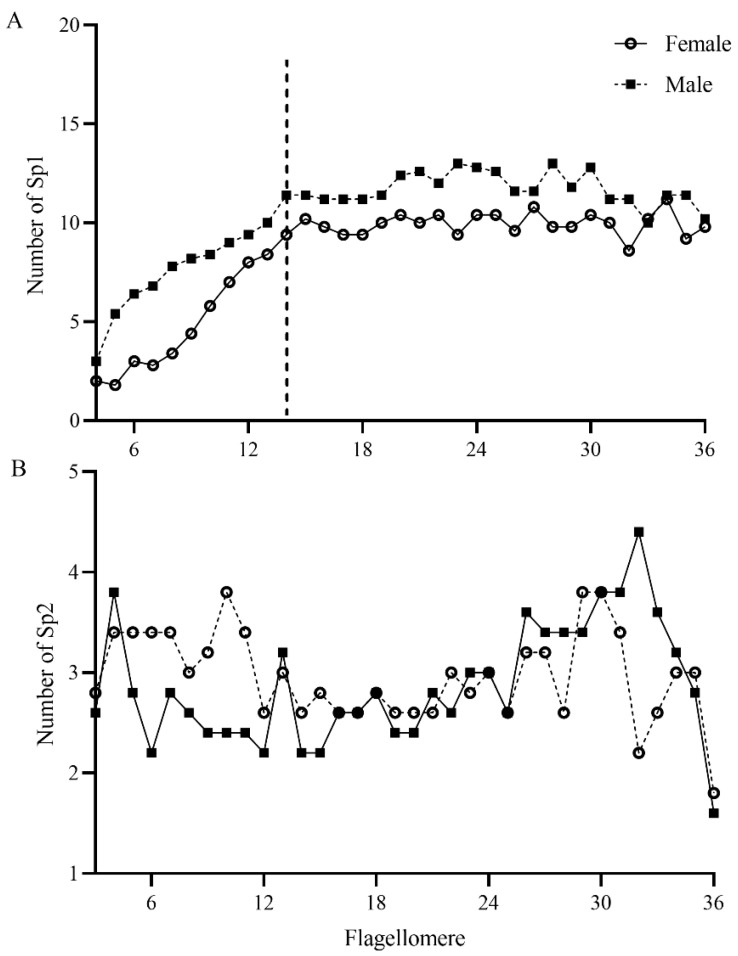
Numbers of sensilla placodea 1 (**A**) and 2 (**B**) on antennal flagellum of both sexes of *C. qinlingensis* (N = 5). As the dashed line shown in (**A**), the number of Sp1 increases gradually from flagellomere 1 to flagellomere 14 (**left**) but is almost the same among other flagellomeres (**right**).

**Figure 10 insects-13-00907-f010:**
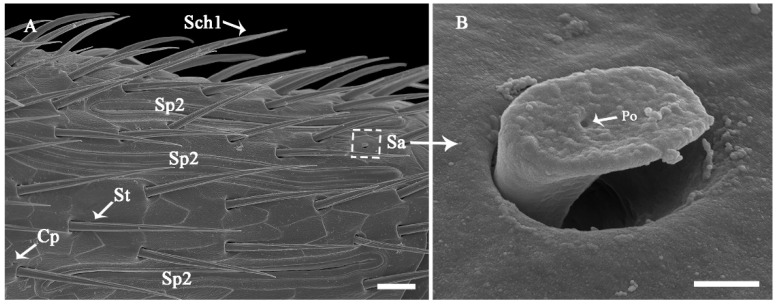
Fine morphology and distribution of antennal sensilla on flagellomere 20 of *C. qinlingensis*. (**A**) Ventral view of flagellomere 20. (**B**) External morphology of sensilla auricillica protruding from a round socket, with a pore clearly visible at the center of the ear-expanded structure. Sa, sensilla auricillica. The same acronyms and abbreviations are used in subsequent figures. Scale bar: (**A**) = 10 μm; (**B**) = 500 nm.

**Figure 11 insects-13-00907-f011:**
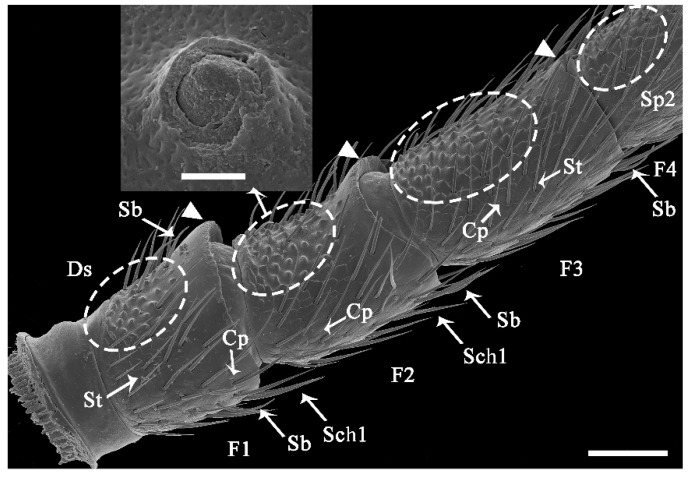
Fine morphology and distribution of antennal sensilla on the basal flagellomere of *C. qinlingensis*. Dotted circles show concentrated distributions of dome-shaped sensilla from a lateroventral view of flagellomeres 1–4. The inset shows thick and multi-layered cuticles perforated by surface pores. Ds, dome-shaped pores. The same acronyms and abbreviations are used in subsequent figures. Scale bar = 50 μm, Inset = 2 μm.

**Figure 12 insects-13-00907-f012:**
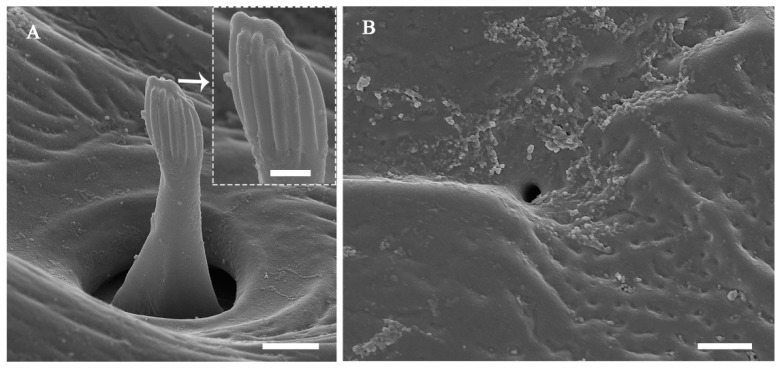
Fine morphology of sensilla coeloconica and cuticular pores on the antennae of *C. qinlingensis*. (**A**) Characteristic peg with finger-like projections located in a flexible cuticular socket on flagellomere 35. The inset shows that no surface pores are present on fingerlike projections. (**B**) Cuticular pore on the antennal cuticle. Scale bar: A and B = 1 μm; inset = 500 nm.

**Figure 13 insects-13-00907-f013:**
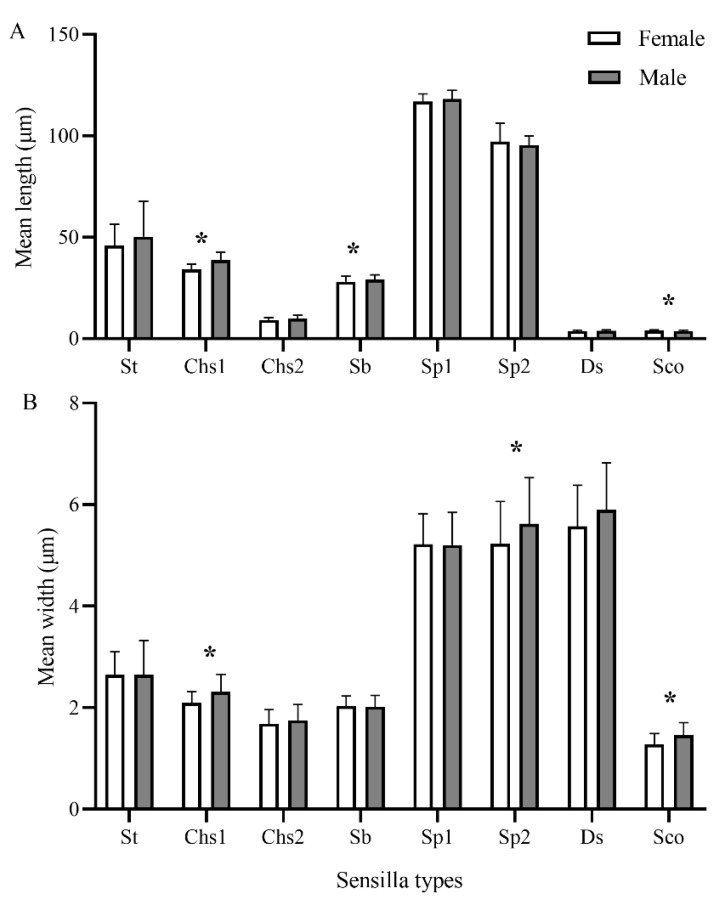
Length and width of different types of antennal sensilla in both sexes of *C. qinlingensis*. Data are the mean ± SD of at least 40 individual sensilla of five wasps of each sex. Asterisks (*) indicate significant differences between the two sexes (*p* < 0.05).

**Table 1 insects-13-00907-t001:** Mean size of the antennal segments in female and male *Coeloides qinlingensis*.

Antennal Segment	Sex	Length	Width
Scape *	Female	277.55 ± 30.75	136.35 ± 17.38
	Male	280.51 ± 22.24	149.46 ± 11.26
Pedicel	Female	92.45 ± 10.15	107.95 ± 11.47
	Male	95.53 ± 7.49	114.31 ± 8.56
Flagellomere	Female	115.75 ± 10.87 b	68.35 ± 8.61
	Male	123.79 ± 14.66 a	74.42 ± 5.66
Total	Female	4406.43 ± 317.18 b	
	Male	4817.69 ± 139.98 a	

* Data are the mean ± SD. Scape length includes the radicula. Different letters in the same column indicate significant differences between female and male wasps.

**Table 2 insects-13-00907-t002:** Morphological characteristics and distribution of the antennal sensilla of female and male *Coeloides qinlingensis*.

Sensilla	Tip	Shape	Wall	Porosity	Socket	Length (μm)	Diameter (μm)	Distribution
St	Sharp	Slightly curved	Grooved	Aporous	Flexible	25.25–89.03	2.55–6.86	All
Sch1	Sharp	Straight	Grooved	Aporous	Flexible	28.60–45.23	1.55–3.50	All
Sch2	Sharp	Straight	Smooth	Aporous	Flexible	7.32–15.19	1.17–2.55	Sc, Pe
Sb	Blunt	Straight	Grooved	Uniporous	Flexible	23.56–35.35	2.07–4.10	Fl
Sp1	Blunt	Straight	Smooth	Multiporous	Inflexible	108.99–127.25	3.87–6.91	F3–F36
Sp2	Blunt	Straight	Smooth	Multiporous	Inflexible	83.15–119.51	3.80–7.68	F3–F36
Ds	Blunt	Straight	Grooved	Multiporous	Inflexible	2.60–4.96	3.74–8.63	F1–F4
Sa	Blunt	ear-shaped	Coarse	Uniporous	Flexible	0.40–1.73	0.33–0.87	Odd F15–F33
Sco	Blunt	Straight	Grooved	Aporous	Flexible	2.76–4.81	2.50–4.22	Even F15–F33
Cp		Small hole					0.41–0.90	Fl

**Table 3 insects-13-00907-t003:** Number and distribution of different types of sensilla on the antennae of female and male *C. qinlingensis*.

Sensilla Type	Sex	Scape	Pedicel	Flagellum	Total
St	Female	32.6 ± 7.3	35.0 ± 1.6	3982.0 ± 91.0 b	4049.6 ± 95.7 b
	Male	42.4 ± 3.6	37.0 ± 1.0	4772.4 ± 226.0 a	4851.8 ± 227.2 a
Sch1	Female	15.0 ± 3.5	7.2 ± 1.6	293.2 ± 18.5 b	315.4 ± 19.2 b
	Male	10.8 ± 1.1	7.2 ± 1.1	357.6 ± 16.9 a	375.6 ± 17.7 a
Sch2	Female	87.8 ± 9.9	8.4 ± 2.1		96.2 ± 10.2
	Male	87.2 ± 10.1	7.8 ± 1.1		95.0 ± 9.2
Sb	Female			178.2 ± 4.3	178.2 ± 4.3
	Male			200.2 ± 24. 8	200.2 ± 24. 8
Sp1	Female			261.4 ± 24.3 b	261.4 ± 24.3 b
	Male			337.4 ± 21.7 a	337.4 ± 21.7 a
Sp2	Female			107.0 ± 25.0	107.0 ± 25.0
	Male			93.6 ± 16.4	93.6 ± 16.4
Ds	Female			183.8 ± 13.8 b	183.8 ± 13.8 b
	Male			274.8 ± 20.9 a	274.8 ± 20.9 a
Sa	Female			9.8 ± 2.1	9.8 ± 2.1
	Male			10.4 ± 2.6	10.4 ± 2.6
Cos	Female			13.8 ± 3.3	13.8 ± 3.3
	Male			14.2 ± 1.3	14.2 ± 1.3
Cp	Female			125.0 ± 23.2	125.0 ± 23.2
	Male			142.6 ± 21.5	142.6 ± 21.5

Data are the mean ± SD of at least 40 individual sensilla of five wasps of each sex. Different letters in the same column indicate significant differences between female and male wasps (*p* < 0.05).

**Table 4 insects-13-00907-t004:** Features of the sensilla placodea of the antennae of female and male *C. qinlingensis* (N = 5).

Sensilla	Sex	Density	Surface Pores (ind./μm^2^)	Total Surface Area (μm^2^)
Sp1	Female	10 ± 1	7.43 ± 1.03 b	1059.53 ± 119.13 b
	Male	9 ± 1	12.23 ± 1.54 a	1158.38 ± 114.63 a
Sp2	Female	2 ± 1	12.85 ± 1.12 b	1158.38 ± 114.63 b
	Male	2 ± 1	41.10 ± 2.10 a	825.50 ± 95.80 a

Data are the mean ± SD. Different letters in the same column indicate significant differences between female and male wasps (*p* < 0.05).

## Data Availability

The data presented in this study are available in the article.

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
