# Peer review of "Functional Morphology of the Antennae and Sensilla of Coeloides qinlingensis Dang et Yang (Hymenoptera: Braconidae)"

_insects, 2022, doi:10.3390/insects13100907_

Round 1
Reviewer 1 Report
In the manuscript, the authors examined the fine morphology of the antennae of Coeloides qinglingensis, as well as the type, shape, and distribution of antennal sensilla. The manuscript is clearly structured. I have only some small suggestions for the manuscript.
Can you provide some pictures showed the morphology of adults of C. qinglingensis?
Line 13: the larval parasitoid Coeloides qinglingensis, larvae or adults, which one?
Line 313: The sentence “whether St, Sch1 or Sch2, these had no significant differences” needs to revise.
Line 314: What does “significantly greater” mean?
Line 544: The parentheses is needless.
In Table 2: Please check the different letters following the same column.
In Figure 12A, if there are significant differences of Sb and Sco between bothe sexes, the differences may also be in the remaining sensilla between bothe sexes. Please check the raw data.
In the tables and figures, you should point out the sample number and standard deviation or standard error when you conducted variation analysis.
Author Response
Dear Editor and reviewers of Insects:
Thanks for your very informative revision suggestions to our manuscript (insects-1944089). We especially appreciate three anonymous reviewers who have carefully read the manuscript and given the authors crucial advice on how to improve the manuscript. Reasons we will explain why in every reviewer response. We entirely accept his/her suggestions. All modifications were marked using the “Track Changes” function in MS word.
Furthermore, we have replaced the Cui-Hong Yang’s email and redefined the Zhi-Xiang Liu’s contribution. All authors have signed these contribution forms. On the other hand, one English-speaking natives, Dr Chris Akcali, have polished language errors in the revised manuscript. We suggest that the revised manuscript has reached the aim of your esteemed journal, and please check it.
We reclaimed that this manuscript has not been published or presented elsewhere in part or entirety and is not under consideration by another journal. All the authors have approved the manuscript and agree with its submission to the Insects journal. There are no conflicts of interest to declare.
Yours sincerely
Zong-Bo Li, PhD.
Southwest Forestry University,
300 Bailongsi, 650224,
Kunming, Yunnan, China
Correspondence: [email protected]

Reviewer 2 Report
Dear Authors,
The work is very well written, and I have no substantive comments.
I think two things could be improved:
1) Keywords should not duplicate the words used in the title. May be used: parasitic wasp, sensory organs, ultramorphology.
2) The full name of all species should be provided, i.e. together with the year of the species description. The papers containing the original descriptions should be mentioned in the references. This is a very important rule that should be used in all work.
Best wishes
Author Response
Thanks for your very informative revision suggestions to our manuscript (insects-1944089). We especially appreciate the reviewers who have carefully read the manuscript and given the authors crucial advice on how to improve the manuscript. Reasons we will explain why in every reviewer response. We entirely accept his/her suggestions. All modifications were marked using the “Track Changes” function in MS word.

Reviewer 3 Report
I suggest doing a mapping of each kind of sensilla for each antennomere. You can use as model the fig 2 from the paper “Between extreme simplification and ideal optimization: antennal sensilla morphology of miniaturized Megaphragma wasps (Hymenoptera: Trichogrammatidae)” (https://peerj.com/articles/6005/).

Author Response

(The authors gave the same response as above.)
